# Using Reciprocal Transplants to Assess Local Adaptation, Genetic Rescue, and Sexual Selection in Newly Established Populations

**DOI:** 10.3390/genes12010005

**Published:** 2020-12-23

**Authors:** Jacques Labonne, Aurélie Manicki, Louise Chevalier, Marin Tétillon, François Guéraud, Andrew P. Hendry

**Affiliations:** 1Université de Pau et des Pays de l’Adour, UMR INRAE-UPPA, Ecobiop, FR-64310 Saint-Pée sur Nivelle, France; aurelie.manicki@inrae.fr (A.M.); louise.chevalier@inrae.fr (L.C.); marin.tetillon@inra.fr (M.T.); francois.gueraud@inrae.fr (F.G.); 2Redpath Museum and Department of Biology, McGill University, Montreal, QC H3A 0C4, Canada

**Keywords:** genetic rescue, local adaptation, mating success, gene flow, small population

## Abstract

Small populations establishing on colonization fronts have to adapt to novel environments with limited genetic variation. The pace at which they can adapt, and the influence of genetic variation on their success, are key questions for understanding intraspecific diversity. To investigate these topics, we performed a reciprocal transplant experiment between two recently founded populations of brown trout in the sub-Antarctic Kerguelen Islands. Using individual tagging and genetic assignment methods, we tracked the fitness of local and foreign individuals, as well as the fitness of their offspring over two generations. In both populations, although not to the same extent, gene flow occurred between local and foreign gene pools. In both cases, however, we failed to detect obvious footprints of local adaptation (which should limit gene flow) and only weak support for genetic rescue (which should enhance gene flow). In the population where gene flow from foreign individuals was low, no clear differences were observed between the fitness of local, foreign, and F1 hybrid individuals. In the population where gene flow was high, foreign individuals were successful due to high mating success rather than high survival, and F1 hybrids had the same fitness as pure local offspring. These results suggest the importance of considering sexual selection, rather than just local adaptation and genetic rescue, when evaluating the determinants of success in small and recently founded populations.

## 1. Introduction

Local adaptation (LA) happens when individuals have higher fitness in their local environment than do immigrant individuals [1]. LA is built via selective processes, wherein some individuals achieve higher survival and reproductive success than others. The genetic contribution of these individuals is therefore more likely to be passed on to the next generation, further shaping the new identity of the local gene pool and their phenotypic traits [2,3,4]. In some cases, however, LA can be compromised by limited genetic variation, especially in small populations and/or on colonization fronts (i.e., the margins of a distribution area where individuals are colonizing new habitats). In such cases, the effects of genetic drift can counteract the efficacy of LA [5]. But moderate gene flow sometimes can improve population fitness [6,7,8] and even rescue it from extinction (“genetic rescue” or GR, ref. [9,10,11,12]), which then enhance subsequent LA. On the other hand, gene flow can introduce non-adapted alleles into the population, increasing the risk of severe maladaptation that can lead to extinction [13,14].

In the context of the sixth biodiversity crisis [15], documenting the interaction between LA and GR is of major importance for understanding how organisms might cope with rapid environmental change in fluctuating demographic contexts [16]. That is, many species and populations that exist in a tenuous demographic state, such as low population size, must now also face a rapidly changing environment. Moreover, many species are shifting their ranges and colonizing new environments, either naturally as a response to changing environments or unnaturally through human-mediated introductions. In this complicated intersection of colonization, demography, and environmental change, rapid LA to new environmental conditions becomes critical [17]. It thus appears paramount to understand the speed at which LA arises, and whether or not GR is important, when organisms colonize (or experience) new environments.

The pace of LA can be investigated through the study of fitness across generations following the migration of a pool of individuals into a new environment. Provided founding genetic variation is not limiting, efficient LA by selection should rapidly improve the fitness of residents. After that period of rapid adaptation by residents, any new immigrants should incur a fitness disadvantage in terms of survival, reducing their odds of transmitting their alleles to the next generation, compared to locally adapted resident individuals. If the immigrants and residents interbred, LA should then translate in a lower overall fitness for hybrids compared to local individuals. By contrast, GR would be expected to be evident as an increase in hybrid fitness, though the benefits of increased genetic variation. This latter signature of GR is especially expected in the context of small populations on colonization fronts, wherein standing genetic variation can be reduced through potent genetic drift and where inbreeding can drastically impact the fitness of local individuals. Further, in each generation, fitness differences can be decomposed into mating success (sexual selection) and survival (natural selection), with the latter being more directly indicative of adaptive effects. These components of fitness can be related to the degree of mixture between local and foreign genes (Hybrid index, [12]).

Salmonid fishes have contributed actively to our understanding of LA [18,19,20,21,22], with strong evidence of LA being found in many populations of various species [23,24]. The mechanisms underlying these adaptations have been investigated through correlational approaches [25,26,27,28], genomic analyses [29,30], common garden experiments [31], and some reciprocal transplant experiments [32]. For most of these studies, however, the sampled populations were already established, and at a (presumably) stable equilibrium. As a result, we have very little understanding of the rate and determinants of LA in its earliest stage. One exception to this research gap is the study of LA in invasive brown trout in Newfoundland, wherein local—but recent—populations fared better in terms of survival compared to foreign introduced populations [33]. However, LA is not just about survival differences—but also about mating and reproductive success. Further, LA can also influence the success of hybrids and backcrosses. Hence, we also need studies of LA in new populations conducted on a multigenerational scale [22]. GR has been less well studied in salmonids, although some studies suggest that small isolated populations have very low genetic variation, and might therefore benefit from gene flow [34,35].

We studied how these processes played out following an introduction of brown trout (*Salmo trutta* L.) to the remote sub-Antarctic islands of Kerguelen [36]. Of particular interest were populations on the western side the colonization front [37], because these populations face increasingly challenging environmental conditions due to their close proximity to the melting ice cap [38,39]. Rapid LA might be especially important in such cases [40]. In an earlier study of this system, we investigated the fate of two populations introduced in 1993 from just a few founders, finding very high inbreeding levels and selection against homozygosity up to 2010 [41]. Here, then, we have an interesting intersection of LA (potentially favored in the novel environments) and GR (potentially favored owing to low genetic variation and inbreeding). Thus, in 2010, approximately 3 to 4 generations after their initial introduction, we conducted a reciprocal transplant between the two populations [42,43]. If important LA occurred on such a short time scale, introduced foreign individuals should show low survival and reproductive success—resulting in low gene flow (sensu [44]). If GR is important, however, hybrid offspring should be more successful than pure resident offspring, thus enhancing gene flow.

We monitored the fate of this experiment by sampling the two populations again in 2011, 2012, and 2018 to assesses the level and structure of gene flow, and to estimate the fitness of local, foreign, and potential F1, F2 and backcrosses hybrid individuals. In addition, we investigated whether the degree of hybridization (hybrid index) was related to possible components of selection, such as recapture proportions between local and foreign individuals following the transplantation protocol (indicative of adult survival), sired family size (indicative of offspring survival), or homozygosity level (as indicative of inbreeding load).

## 2. Materials and Methods

### 2.1. Sites and Populations Description

Two populations inhabiting different environments (henceforth “systems”) were selected based on our knowledge of their recent past and introduction conditions [36,41]. The Val Travers system, located in the northern area of the main island of Kerguelen sub-Antartic archipelago (491,832″ S, 692,579″ E) is 9 km long, with a gradient of habitats and slopes from mountain to lowland landscapes. It empties into Lake Bontemps (700 ha), which is connected to the sea by a steep outlet. The Clarée system is 3 km long, and is located to the south of the main island (492,935″ S, 693,744″ E) on a plain featuring several interconnected arms originating from Lake Hermance (350 ha) and also from a tributary flowing from a nearby glacier (River Galets). The Clarée empties directly into a shallow marine bay.

The trout populations in both systems were artificially introduced in 1993 from two other Kerguelen systems [36]. The Val Travers population was founded with 2000 six months old juveniles from a single cross between one male and one female from the River Chateau, which was first colonized in 1962. The Clarée population was founded with 1700 six months old juveniles from a cross between one female and two males that were captured while migrating upstream in the River Armor, which at that time was not yet colonized by brown trout (no natural reproduction observed [36], see Figure 1). The genetics of these populations have been investigated [41], revealing a high initial level of inbreeding in both populations. However, subsequent selection against homozygotes was also detected in the first generations, especially in Val Travers.

### 2.2. Transplantation Experiment and Populations Sampling

In March 2010, in each system, we used electrofishing to sample 261 non-mature individuals aged from 1 two 3 years old (mean body size 13.55 cm for Val Travers, 11.46 cm for Clarée). Each fish was anaesthetized using phenoxy-ethanol, measured for body size, weighed, and individually tagged with PIT-tags. For each fish, we clipped a piece of caudal fin, and placed the clip in 96% ethanol for further genetic analysis. Fish were then placed in submerged cages for 14 days (3 cages per population) to ensure recovery from handling: one fish died in Val Travers, and three died in Clarée over this period. In each population, the surviving individuals were separated in two lots. The first lots (107 fish for Clarée, 109 fish for Val Travers) were released on site, so to have a proxy of resident (or local) fish survival through recapture in each population for the next years (2011 and 2012). The second lots (151 fish for both populations) were immediately transported by helicopter (15 min travelling time) to the other system (Val Travers origin to Clarée, and vice versa), where they were released as foreign individuals. We will refer to all these fish as cohort 0 (C0) hereafter, encompassing resident fish (“local”) and transplanted fish (“foreign”).

In three designated areas for each system, a sampling protocol using electrofishing (2-pass depletion method on a fixed area) was applied to estimate local densities (in 2010, 2011, and 2012, see Appendix A), and to potentially recapture C0 tagged fish (in 2011 and 2012). In 2012 and in 2018, we also sampled both systems specifically for young-of-the-year offspring (approximately 6 months old fish) at the same sites that were previously used for density estimation. We also extended these latter samplings to stretches of river between these sites, to minimize the risk of over-representing foreign or local parental contribution. These offspring were anaesthetized, then killed with an overdose of anesthetic, and kept in 96% ethanol for further genotyping. The 2012 sampling was conducted to detect the first potential F1 hybrid offspring between local and foreign parents from C0 (since transplanted individual ages ranged from 1+ to 3+ two years before, and most individual start reproducing at 5 years old in Kerguelen Is., [45]). These offspring will be referred as to C1 hereafter. The 2018 sampling was conducted to potentially detect not only F1 individuals, but also F2 of either local or foreign origin, and backcrosses with either local or foreign individuals. These offspring will be referred as to C2 hereafter.

### 2.3. Ethical Statement

At the time of the transplantation experiment (2010), no ethical committee was constituted and recognized in France. All procedures however were previously submitted to the scrutiny of the French Polar Institute as well as the Natural Reserve of the French sub Antarctic islands for evaluation, and were approved. For the 2018 sampling, authorization APAFIS#16249-201807241223324 was delivered by the French committee for ethics in animal experimentation *n*°073.

### 2.4. Genetic Analyses

To estimate the potential gene flow in each population after transplantation, we genotyped C0, C1, and C2 individuals using 15 microsatellite markers (see Appendix B for details regarding DNA extraction, markers amplification and genotyping methods, Appendix B
Table A1 for markers error rates). These markers are located on different linkage groups [46], and provide satisfactory discriminant power to contrast the two populations (FST = 0.1426). The number of fish genotyped per population and per year is shown in Appendix B.

### 2.5. Genotypic Categories Assignation and Reconstruction of Families

We used the NewHybrids 1.1 software [47] to reconstruct the structure of gene flow in our transplantation experiment for each population. In essence, NewHybrids attempts to assign individuals to a mixture of genotypic categories representing the possible structure of the gene flow. Because the C1 samples could be only either from pure origin (local or foreign) or F1 hybrids, we ran a first analysis using only C0 genotypes and C1 genotypes. C0 genotypes contributed to improve allelic frequencies estimation but were not used to estimate the π mixture, which was only assessed using C1 offspring genotypes (see [47]). Using these first assignments for the C1 offspring, we then ran a second analysis, integrating this time the C2 offspring samples, so as to determine their own specific π mixture, and to assign them to all possible genetic classes (pure local, pure foreign, F1, F2, backcross with local, backcross with foreign). This two-step analysis approach allowed us to better reflect the transplantation protocol and to benefit from our precise knowledge of the possible genetic categories that could potentially be found in 2012 and 2018, respectively. For each population, the analysis was realized using all samples genotyped on a minimum of 10 microsatellites. After 10,000 iterations for burning, 100,000 iterations were run to estimate the model’s parameters on three different MCMC chains. We checked the stability of the estimates by running 3 different chains: the average difference in individual assignation probabilities to the various genotypic categories was 0.0003189 and 0.0001413 for Val Travers and Clarée, respectively.

To delineate families among C1 and C2 gene pools, an analysis was run separately in each river using COLONY 2.0.6.5 software [48]. In particular, COLONY uses multi-locus genotypes to infer sibship among samples. Potential parental genotypes were included in the analyses (84 females and 116 males for Val Travers, 51 males and 67 females for Clarée). We performed long runs, using weak priors and full likelihood method, assuming polygamy for males and females, with inbreeding, for diploid dioecious species. These tests were repeated three times to validate results manually. We then tagged families as either local, foreign, F1, F2 or backcrosses by simply matching the individual assignations obtained from NewHybrids with the family structure obtained from COLONY. In some cases, some families could not safely be assigned because they were composed of more than one type of offspring (for instance, both local individual and hybrid individual were found in the same family, for a same run of the analysis, or between runs). These families were not used in the following analyses (they represented 5.4% of families for Val Travers and 5.65% for Clarée).

All data files and additional settings for NewHybrids and Colony softwares are accessible online (https://doi.org/10.15454/NDFQJD).

### 2.6. Estimating Fitness

Our general approach to estimate relative fitness was to calculate the genetic contribution of the different genotypic categories of C0 individuals (local, foreign) to the C1 gene pool, and then the contribution the different genotypic categories of C1 individuals (local, F1, foreign) to the C2 gene pool. The approach is straightforward for C1 individuals: the data describing genotyping categories contain both C1 and C2 genotypic frequencies.

For C0 individuals, however, the initial proportions of transplanted (foreign) and resident (local) C0 individuals, in relationship with our field sampling protocol of C1 individuals, are not perfectly known. For instance, if foreign individuals move far from their release site and reproduce out of our sampling area, we might underestimate their total contribution to the next generation. To account for this, we here envisioned two different scenarios. In a first scenario, we assumed restricted dispersal, wherein foreign individuals would not move too far from their release site (and therefore from our sampling sites). To do so, we accounted for the usual home range known for brown trout, wherein most individuals remain within a 300 to 500 m linear of the river [49,50]. We multiplied this length by the average width of each river to obtain the surface area (which yielded about one hectare in both systems). We then calculated the likely proportion of foreign individuals by dividing the number of transplanted individuals by the total number of individuals expected on the surface area. The second scenario, however, assumed that individuals could move in the whole system, making the likelihood for them to sire offspring in our sampling area much smaller. The surface area in this scenario was therefore much bigger, and accounted for all habitable area for each population. Using these two scenarios enabled us to account for uncertainty in the proportion of transplanted individuals among potential parents in our sample, each scenario representing an extreme situation for dispersal or sampling bias regarding transplanted individuals. This uncertainty is thus accounted for in the calculation of C0 individuals’ fitness.

To test whether fitness was different between the genotypic categories (for C0 and C1 individuals), we compared the observed genetic contribution to the expected genetic contribution assuming full random association between gametes, and equal survival and capture probabilities among offspring up to sampling date. We used Goodness-of-Fit X^2^ tests to assess the statistical significance of differences between observed and expected genetic contributions.

### 2.7. Components of Selection

We looked at different components of selection during the transplantation experiments. First, we compared recaptured proportions between local and foreign C0 individuals, between 2010 and 2012, using the Fisher exact-test, as a proxy of their respective survival until the first potential reproduction.

We then ranked all individuals and families according to their hybrid index ([12]: 0 for local, 0.25 for F1xlocal, 0.5 for F1 and F2, 0.75 for F1xforeign, and 1 for foreign). For C1 and C2 gene pools, we tested whether family size (a proxy of survival between birth and sampling date, at 6 months old) was related to the hybrid index, using a polynomial model with Gaussian distributed error. The polynomial approach allows to detect linear and non-linear relationships between the hybrid index and the variable of interest. The statistical significance of linear and non-linear components of the model was assessed using F-tests on variance ratios.

Finally, because the two studied populations have been founded by very small numbers of parents and because selection against homozygotes was shown to be active [41], we assessed the Homozygosity Level of each individual (HL, [51]) using the Rhh package in R [52]. For all individuals (C0, C1 and C gene pools), we tested whether HL was related to the hybrid index, using again a polynomial model with Gaussian distributed error, and F-tests on variance ratios.

## 3. Results

### 3.1. Gene Flow

Through NewHybrids assignment, we determined the most probable genotypic categories in both populations for C1 individuals and then C2 individuals (Figure 2). In Val Travers, among the 432 C1 individuals sampled, 72.6% were pure local, 2.3% were pure foreign, and 25% were F1 hybrids. In Clarée, among 528 C1 individuals, 99.6% were pure local, none were foreign, and only 0.4% individuals were assigned as F1 hybrids. These results indicate that, in both populations, foreign transplanted individuals achieved some mating success, although with contrasting efficiencies. For the C2 individuals in Val Travers (*N* = 236), 42.8% were assigned as pure locals, 15.7% as F1 Hybrids, 5.5% as F2 Hybrids, and 33.5% and 2.5% as backcrosses with local and foreign categories, respectively. In Clarée, among 183 individuals sampled, 92.3% were pure local, and 7.7% were assigned as backcrosses with the local category. No pure foreign individuals were detected in Clarée nor Val Travers in the C2 gene pool. To sum up, gene flow occurred in both populations, although not to the same extent: whereas non pure local individuals represented 2.64% of the population in Clarée overall, they amounted to 37.87% in Val Travers.

### 3.2. Fitness of the Different Genotypic Categories

To calculate the fitness of the C0 individuals from the different genetic groups, we first estimated the proportion of the foreign (transplanted) individuals in the populations relative to local individuals. To do so, we first estimated local densities to 2161 and 1475 individuals per hectare in Val Travers and Clarée, respectively (Appendix A and Appendix C). Under a restricted dispersal scenario, wherein the 151 foreign transplanted individuals would remain close from their release location, we estimated that they represented 6.98% and 10.24% of the sampled populations for Val Travers and Clarée, respectively (Appendix C, Table A2, Table A3, Table A4 and Table A5). Under this scenario, the genetic contribution of foreign C0 individuals was two-fold higher than expected in Val Travers (X*^2^*
_1 df_, *p* < 0.0001, Table 1). On the contrary, the genetic contribution of foreign C0 individuals in Clarée was 36 times less than expected under random association of gametes (*p* < 0.00001).

When we relaxed the restricted dispersal assumption, assuming the foreign transplanted individuals could disperse in the whole system, the proportions of the foreign transplanted individuals were, respectively, 0.77% and 0.57% for Val Travers and Clarée (Appendix C, Table A2, Table A3, Table A4 and Table A5). Under this scenario, the genetic contribution of foreign C0 individuals was 20 times higher than expected in Val Travers (X*^2^*
_1 df_, *p* < 0.0001, Table 1), whereas it was 2 times lower than expected in Clarée, although this latter difference was not statistically significant (*p* = 0.08).

Therefore, foreign individuals clearly had a better fitness than local ones in Val Travers whatever the dispersal scenario considered. In Clarée, foreign individuals had a lower fitness under a restricted movement scenario, or were on par with local individuals under an unrestricted dispersal scenario.

C1 individuals, the observed genetic contribution of the two origins (foreign and local), differed from random expectation in Val Travers, with most of the deviation due to a greater contribution of foreign individuals, who produced 4 times more offspring than expected, whereas F1 individuals appeared to perform similarly to local individuals (X*^2^*
_2 df_, *p* < 0.0001, Table 2). In Clarée, F1 hybrids appeared to outperform local individuals, producing 7 times more offspring than expected (X*^2^*
_1 df_, *p* < 0.0001, Table 2).

### 3.3. Components of Selection

We first considered potential difference in survival among C0 individuals after release. In Val Travers, 32 out of 109 local fish (29.36%) were recaptured, whereas only 1 out of 151 foreign fish (0.66%) was recaptured, suggesting a considerable apparent disadvantage for foreign individuals (Fisher’s exact test, *p* < 0.00001). In Clarée, 5 out of 107 local fish (4.67%) were recaptured, and 7 out of 151 foreign fish (4.63%) were recaptured (Fisher’s exact test, *p* = 1), indicating no apparent disadvantage for foreign individuals.

Investigating whether early family size (a proxy for offspring survival) could be related to Hybrid Index, we found no significant relationship between both variables in Val Travers (*p* = 0.5424 for the linear term, *p* = 0.2898 for the non-linear term) nor in Clarée (*p* = 0.7906 for the linear term, *p* = 0.3385 for the non-linear term, Table 3). We found a significant relationship between Homozygosity Level (HL) and Hybrid Index (Figure 3, Table 4) in Val Travers (*p* < 0.0001 for the linear term, *p* < 0.0001 for the non-linear term) and in Clarée (*p* = 0.0231 for the linear term, *p* = 0.0337 for the non-linear term): intermediate Hybrid Index individuals appeared to have lower HL values than extreme Hybrid Index individuals.

Additionally, when investigating the relationship between individual homozygosity HL and family size, we found that individuals originating from larger families had lower values of HL in Val Travers (*p* < 0.0001) and in Clarée, (*p* < 0.0001).

## 4. Discussion

Our study incorporated several key aspects that are usually hard to assemble in a single experiment: (1) both populations were founded at known dates by known numbers of individuals of known ages, (2) the transplantation experiment and our marker set were designed to efficiently distinguish the various genotypic categories over two generations, and, to some extent, (3) we could verify whether differences in fitness were related to particular life cycle stages or to genetic variation (heterozygosity). Of particular interest was the context of the recent foundations of our populations, which allowed us to simultaneously assess the speed at which LA arose and the potential role of GR. Beyond the usual complexities of reciprocal transplantation experiments [53], we also had to account for uncertainty in proportion of transplanted individuals relative to resident individuals. By envisioning two contrasting extreme scenarios for these proportions, we were able to explore the probable range of fitness differences for first generation foreign and local individuals. We also assessed the fitness of the second generation foreign, hybrid and local individuals. Our results are generally nuanced, with little support for LA, some hints that GR might be operating, and intriguing evidence that variation in mating success (i.e., sexual selection) was a key factor moderating gene flow.

### 4.1. Local Adaptation

LA is a widespread phenomenon in wild populations [44] although it is not always evident [54]. The speed at which such LA evolves is an active research topic, with some examples of substantial LA evolving in fewer than 10 generations [3,18,22]. In our study system, up to 4 generations had passed since the foundation of both populations, a length of time during which at least some selection has been at work. For instance, our previous work showed that selection against homozygosity was clearly active, especially in Val Travers—which had been founded by the progeny of only two individuals. This form of selection was further confirmed in the present study, wherein we detected selection against homozygotes during early life, in both populations. Given ecological differences between their habitats and phenotypic divergence between the populations [55], LA was a logical candidate contributing to such selection. Indeed, evidence for LA evolving on such time scales has been reported for other salmonid systems [56,57,58].

Surprisingly, then, our experiment failed to observe clear footprints of LA that would have reduced gene flow after the transplantation. In the Clarée population, despite observing the same recapture rates between local and introduced foreign individuals, the genetic contribution of foreign individuals was potentially lower than that of local individuals. However, that result hinged on assumptions regarding dispersal of individuals in the system relative to our sampling efficiency. Additionally, although F1 hybrids were few in Claree, they had higher than expected fitness, indicating—at the least—they suffered no selective disadvantage overall. In Val Travers, we recaptured significantly fewer transplanted foreign individuals that expected. This difference could reflect LA, adaptive plasticity in the early stages of life (before the transplant took place), or a lack of local experience. The low recaptures rate of these individuals also could be related to behavioral response to transplantation, if transplanted individuals were more likely to disperse [35,59]. However, beyond recaptures of the transplanted individuals, most of our data suggested a lack of LA. In particular, transplanted foreign individuals seemingly had very high mating success, and the fitness of F1 hybrids was similar to that of local individuals. For instance, we did not find footprints of differential survival between local or hybrid individuals (a finding also present in [32]).

How can we explain this weak (if any) LA between our study populations? One possible explanation is that the spatial and temporal scale of LA is larger than the contrast examined in our experiment [22]. Indeed, the founding individuals all originated from the Kerguelen islands where the species was introduced and first reproduced naturally in 1962 (although they probably do not stem from the main strain in Europa though, possibly boosting available genetic variation through admixture, see [36]). Thus, prior to colonizing Val Travers and Claree, brown trout had been subject to about 10 generations of selection in these sub-Antarctic environments. This strong selection for adaptation to the overall Kerguelen conditions might have been the over-riding determinant of selection—as opposed to the finer-scale adaptation to local (stream-specific) conditions [60]. Another possible explanation is that the very low number of founders of the two study populations did not contain sufficient genetic variation to enable rapid LA—at least not within the 4 generations that we studied following establishment. Indeed, the general literature often points at reduced genetic variation as an obstacle for LA in small populations [10,61], which naturally leads us now to the “genetic rescue” hypothesis.

### 4.2. Genetic Rescue

GR had the potential to contribute substantially to the results of our experiment because both populations were strongly inbred, a situation where outbred hybrid offspring could be expected to have higher fitness than inbred resident offspring [10,61,62,63,64]. Although both populations provided tests of GR, the Val Travers population was especially informative due to substantial interbreeding that occurred between local and foreign individuals following the transplantation. Our assessment of GR combined three levels of insight: fitness of hybrids, footprints of selection favoring hybrids, and increased genetic variation in hybrids. Most importantly, the fitness of F1 hybrids was equal to the fitness of pure residents in Val Travers, and only slightly higher than the fitness of pure residents in Clarée, thus indicating weak (if any) footprints of GR. Additionally, we did not find any evidence of increased offspring survival (as assessed via the size of hybrid families relative to that of pure families). This absence of GR is not unprecedented in the literature on salmonids. For instance, Robinson et al. [35] also failed to find significant benefits of outbreeding for family size in brook trout (*Salvelinus fontinalis*). Interestingly, and like Robinson et al. [35], we found some evidence of increased genetic variation as would be expected under GR, wherein individuals with intermediate hybrid index had lower homozygosities. This pattern was especially obvious in Val Travers, possible due to the higher sample size or perhaps because this population had initially very low effective size (Ne = 12, [41]). These patterns of selection could signal ongoing purging of inbreeding load in the population.

We emphasize that our study does not provide unequivocal evidence against the action of GR. First, although we do have demographic data on these two populations, they remain imprecise and it is too early to correctly assess a demographic change at the population scale related to the transplantation. Second, it is also too early to fully assess the fitness of the second generation of hybrid individuals (F2 and backcrosses) beyond finding similar family sizes compared to other genotypic categories. Instead, the benefits of increased genetic variation might become apparent later in the life cycle and might set the stage for further selection and adaptation processes that will perhaps be visible in future generations [10].

### 4.3. Other Drivers of Gene Flow

Having found only weak—if any—support for LA and GR as important drivers shaping patterns of fitness variation in our populations, we are left to suggest additional forces. We first note the different results obtained in the two populations. Clarée seems to conform adequately to a neutral scenario with no evident LA and only weak GR. In Val Travers, however, the fitness of foreign transplanted individuals was 2 to 20 times greater than expected, with the magnitude of the effect depending on the dispersal scenario. This pattern remained in the next generation where foreign individuals showed fitness five times greater than expected. This fitness is the product of two components in our experimental design: the mating success of a genotypic category, multiplied by the survival of the progeny until sampling (6 months-old here). But we demonstrated that family sizes were not different between the different genotypic categories, which implies equal offspring survival between these categories. Variation in fitness can thus be largely attributed to variation in mating success—that is, sexual selection.

These facts indicate that Clarée individuals introduced into the Val Travers may have either superior competitive ability to access potential sexual partners, and/or may be more attractive to local individuals. In general, it is often expected that mating preferences are related to LA, with a preference for locally adapted phenotypes thereby reinforcing the effect of LA on reproductive isolation [65,66,67]. However, this implies that local preference evolved quickly when dealing with newly founded populations on colonization front, a process possibly achieved by runaway selection, but often difficult to observe at the micro-evolutionary scale [68,69,70,71]. Alternatively, and more often, preference for dissimilar phenotype (i.e., inbreeding avoidance, MHC diversity, [72]) may offer a mating advantage to migrants [73] and potentially counterbalance the expected effect of LA. The average body size of introduced Claree individuals was also higher than the body size of Val Travers local individuals among C0 individuals, which could confer a competitive advantage to access sexual partners. However, such advantage should be limited, since they were introduced in an already established population, facing local competitors of higher body size and older ages that were not part of our C0 sampling.

Heterogamous sexual preference is certainly known in salmonids either related to phenotype and origin [74] or to difference in MHC genotypic variation [75,76], but see [77] for negative results in small Atlantic salmon populations. Paralleling that possibility, the effective number of breeders prior to transplantation in Val Travers (Ne = 12) was much lower than that in Clarée (Ne= 46, [41]). Although that difference might partly originate from the founding conditions, the former value is closer to monogamous mating systems expectation than the latter. Such potential difference in mating habits may possibly result in a higher mating success for introduced individuals originating from Clarée. Rare phenotypes can also be sexually favored over more common phenotypes (negative frequency dependent selection, [78,79]). The present nuance here is that it was observed only in one population. Alternatively, the Clarée individuals may bear a trait that would be attractive, notably for Val Travers individuals, but such a trait has to be non-plastic—since the transplanted individuals spent two years in the Val Travers environment before their first possible reproduction.

Finally, assortative mating is known to be context dependent in brown trout: Gauthey et al. [80] have shown that assortative mating was strong when river discharge was not predictable, whereas it could disappear (random mating) when river discharge became very predictable. Whereas the Val Travers watershed features a classic landscape from mountains brooks to a lowland plain, the Clarée system is under the strong influence of the upstream Hermance lake: wind variation on the lake can change the discharge in the river by a factor two or three in a matter of minutes. We correlatively observed that assortative mating with respect to genetic origins was strong in Clarée, a possibly very unpredictable system, whereas it did not occur in Val Travers, the most stable system.

In any case, mating success was probably the key component balancing the gene flow in this experiment: such gene flow could potentially erase any—undetected here—adaptation or founder effects in the next generations. Our finding adds to the growing evidence that sexual selection may have a tremendous effect on evolution [66,81,82,83]. It can possibly promote gene flow towards non-adaptive pathways [84,85,86,87], an outcome that we will endeavor to monitor in the next generations. In particular, changes in conditions could change the relative intensity of the two selection pressures, and upset the equilibrium between the strength of sexual and viability selection [86], shaping patterns of diversity along the colonization range.

## Figures and Tables

**Figure 1 genes-12-00005-f001:**
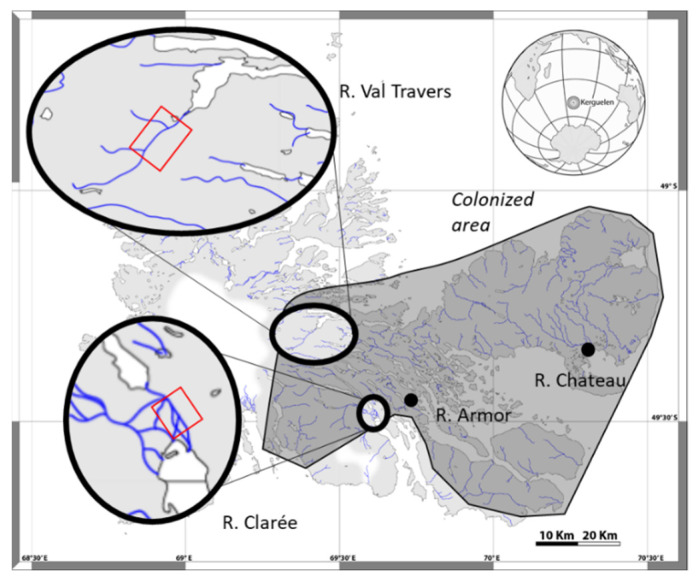
Location of the two studied systems in the Kerguelen islands. In each system, the red rectangle indicates the area where sampling took place. The grey polygon indicates the extent of the colonized area by brown trout in 2018.

**Figure 2 genes-12-00005-f002:**
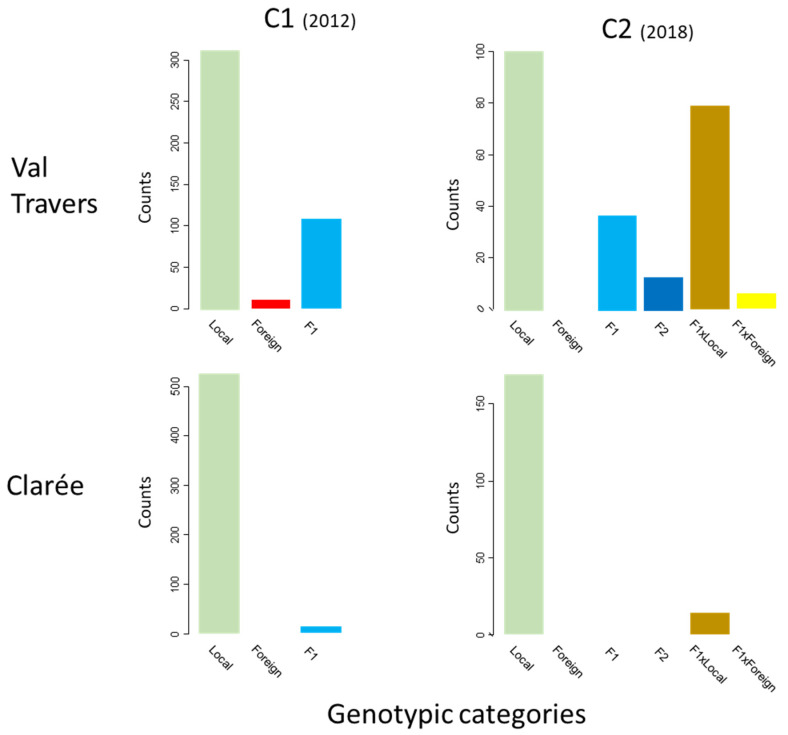
Counts of the different genotypic categories detected for C1 (**left**) and C2 (**right**) gene pools, in Val Travers (**top**) and Clarée (**bottom**) populations.

**Figure 3 genes-12-00005-f003:**
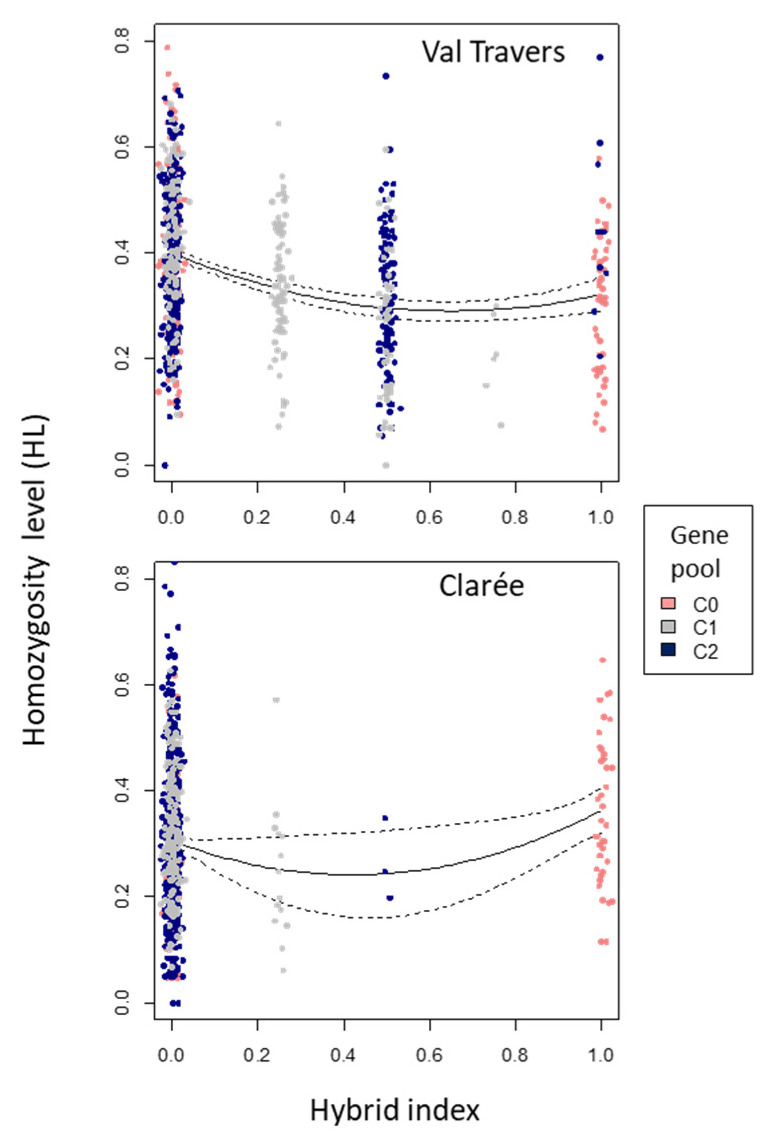
Relationship between Hybrid Index and Homozygosity Level for Val travers (**top** panel) and Clarée (**bottom** panel). The Hybrid index is 0 for local individuals, 0.25 for F1xlocal, 0.5 for F1 and F2, 0.75 for F1xforeign, and 1 for foreign individuals. Red, blue and green dots indicate C0, C1 and C2 individuals respectively. Jittering was added to the X coordinates to improve visibility. The full line represents the polynomial model prediction, the interrupted lines indicating the standard error around this fit.

**Table 1 genes-12-00005-t001:** Goodness-of-fit Tests on the observed fitness (estimate of the number of offspring sired) for the C0 genotypic categories (local, foreign) in Val Travers and Clarée populations, assuming two contrasted dispersal scenarios (unrestricted and restricted) conditioning the initial percentage of transplanted individuals in each population.

Dispersal Scenario.(and Initial Percentage of Transplanted Individuals)	Population	Variables	Genotypic Category		Statistics	
			Local	Foreign	Sum	*p*-Value
		Expected fitness	428.6461	3.3539	432	
(0.77%)	Val Travers	Observed fitness	368	64	432	
		Χ^2^ value	8.580	1096.612	**1105.192**	*p* < 0.0001
**Unrestricted**						
		Expected fitness	524.9973	3.0027	528	
(0.57%)	Clarée	Observed fitness	526.5	1.5	528	
		Χ^2^ value	0.0043	0.7520	**0.7563**	*p* = 0.08
		Expected fitness	410.8147	30.1853	432	
(6.98%)	Val Travers	Observed fitness	368	64	432	
		Χ^2^ value	2.8456	37.8806	**40.7262**	*p* < 0.0001
**Restricted**						
		Expected fitness	473.9511	54.0489	528	
(10.24%)	Clarée	Observed fitness	526.5	1.5	528	
		Χ^2^ value	5.826	51.09	**56.916**	*p* < 0.0001

**Table 2 genes-12-00005-t002:** Goodness-of-fit Tests on the observed fitness (estimate of the number of offspring sired) for the C1 genotypic categories (local, F1, foreign) in Val Travers and Clarée populations.

Population	Variables	Genotypic Category			Statistics	
		Local	F1	Foreign	Sum	*p*-Value
	Expected fitness	171.54	59	5.46	236	
Val Travers	Observed fitness	159	55.5	21.5	236	
	Chi square value	0.916	0.207	47.078	**48.202**	*p* < 0.0001
	Expected fitness	181.96	1.04	0	183	
Clarée	Observed fitness	176	7	0	183	
	Chi square value	0.195	34.165	0	**34.360**	*p* < 0.0001

**Table 3 genes-12-00005-t003:** ANOVA analysis of the linear models testing for the linear and non-linear effects of Hybrid Index on family sizes in Val Travers and Clarée populations.

Population	Hybrid Index Effect	Degrees of Freedom	Mean Square	*F* Value	*p* Value
	Linear term	1	0.064184	0.3717	0.5424
**Val Travers**	Non-Linear term	1	0.194016	0.2898	0.2898
	Residuals	382	0.1726664		
	Linear term	1	0.004937	0.0706	0.7906
**Clarée**	Non-Linear term	1	0.064213	0.9179	0.3385
	Residuals	835	0.069953	1.2004	

**Table 4 genes-12-00005-t004:** ANOVA analysis of the linear models testing for the linear and non-linear effects of Hybrid Index on individual Homozygosity Level in Val Travers and Clarée populations.

Population	Hybrid Index Effect	Degrees of Freedom	Mean Square	*F* Value	*p* Value
	Linear term	1	1.19384	70.456	<0.0001
**Val Travers**	Non-Linear term	1	0.45615	26.921	<0.0001
	Residuals	859	0.01694		
	Linear term	1	0.098087	5.1809	0.02309
**Clarée**	Non-Linear term	1	0.085691	4.5261	0.03367
	Residuals	835	0.018933	1.2004

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
