# Peer review of "Using Reciprocal Transplants to Assess Local Adaptation, Genetic Rescue, and Sexual Selection in Newly Established Populations"

_genes, 2020, doi:10.3390/genes12010005_

Round 1
Reviewer 1 Report
This manuscript is well written and of interest to many readers and audiences.
The title, although accurately describing what occurred in the manuscript, is very vague, and targeted readers may be more likely to find it and make use of it if the words "fish" (generally) or "brown trout" (more specifically) are added between "established populations". Alternatively, one or both of these terms should be included in the keywords.
Line-specific comments/changes/suggestions:
Line 20: put in a long dash (versus the short dash) after "gene flow".
Line 24: change "than" to "as".
Lines 44-45: this is a hard sentence to follow with all the short thoughts separated by commas. Consider revising.
Line 51: first appearance of "GR". Assuming this is genetic rescue, which is mentioned earlier, but not defined as GR.
Line 59: "expansion ranges" is used a couple of times up to this point, and throughout the manuscript, but it is not defined leaving the reader unsure of what is meant by expansion ranges. Please define or explain for clarity.
Lines 70-71: A little confusing with the multiple "therefore" sections... If I understand correctly, the first is saying that LA had already occurred and could not be studied, and the second is saying that the factors causing or affecting LA could also not be studied. Please clarify.
Lines 81-82: "at the boundary of the current western boundary of the expansion range" is confusing. Please clarify.
Lines 127-129: A high level of inbreeding seems counter to selection against homozygotes. Please clarify how both can be occurring in these populations.
Line 139 and throughout: bracket and parentheses change for some parenthetical statements (i.e., here "(" used on one side and "]" used on the other). Please check for and correct throughout.
Line 146 and throughout: note that you refer to supplementary material 1, 2, and 3, but these are labeled and presented as Appendix A, B, and C.
Line 149: anesthetized using what prior to killing, and how were they killed (e.g., pithing, overdose of same chemical used for anesthetization, etc.)?
Lines 151-153: the parenthetical information should be moved up or similar info included in the transplant paragraph above. Were fish in both systems in the 1-3 year range? What was the size of the transplanted fish (i.e., were they the same size at age in both systems or did they differ between systems due to environmental conditions)? If they do differ, would there be a size advantage to survival because foreign fish are larger than local fish or vice versa?
Line 210: this is not a list of the two scenarios. Suggest breaking this into two sentences at the ":".
Line 215: missing ")" after "systems".
Line 233: I think "12" is a citation - put in brackets?
Lines 248-249: This appears to be a part of the template and should be deleted.
Lines 256-260: Is it not possible to obtain a foreign individual in the C2 fish? If not, then delete the category from the Figure 2 C2 panels as it makes it incorrectly seem that these individuals were missing, when in reality, they couldn't exist. If it is possible, but none were caught, then that should be stated in these sentences.
Line 266: capitalize "f" on "frequencies".
Line 266, in Figure 2, and forward: "é" missing in almost all spellings of "Clarée".
Line 271: "(resident)" not needed as local (versus foreign) has already been defined a number of times prior to this point.
Line 276: Is "important" the correct word choice? Should it just be "less than expected"?
Line 290: Define "performed better". In reference to the C0 contribution, are we talking survival, reproduction, etc.? Be more specific as to how they performed better. Looking at Table 1, although foreign fitness was greater than expected, the observed fitness for foreign individuals was still lower than that of local fish. As such, did they perform better better than local fish, or better then they were expected to given a random mating scenario?
Line 309: "considered".
Line 311: I think "apparent may be needed here between "considerable disadvantage", similar to line 313. There could be other reasons these fish were not recaptured.
Lines 321-322: this is presented as something that may not have been expected, but wouldn't this result have been the expected result in F1 generation fish resulting from the spawning of two unrelated populations, especially if they were initially found to be inbred (higher HL)?
Figure 3 caption: Please remind readers what the 0-1 scale on the Hybrid index axis represents (0 local to 1 foreign). Maybe just a repeat of the parenthetical statement on lines 233-234.
Line 352: what is meant by "automatically found"? You can't always find it even when you are looking for it? Or even when it has occurred?
Line 369: "no selective disadvantage overall." Was one expected given that the F1 HL is lower than the C0 fish AND there is selection against hoozygotes?
Line 380: "10 generations of selection in this area". You mean on the island itself prior to the fish being introduced to Val Travers and Clarée in 1993. Otherwise the populations have undergone 4 generations in the two systems, correct?
Line 398: brook trout (Salvelinus fontinalis)
Line 405: "the jury is still out" not often seen in scientific paper.
Line 435-437: Couldn't these Ne also be related to how the two populations were founded (full siblings in Val Travers and full and half siblings in Clarée)?
Line 444: write out authors names and put "80" in [].
Appendix B could use some English language clean up, especially for past and present tenses, spelling out numbers when they start a sentence, and the exclusion of certain connector words to make a complete sentence. In section B), percentages use "." in the paragraph, but "," in the tables. Please correct for consistency. Also, sections C) and D) and their tables are not needed as the information in them is adequately summarized in lines 563-567.
Appendix C, Table A) Why number of fish per square meter? this is very hard to interpret and understand. Could these be standardized to per hectare, or linearly per km? Then at least the numbers would represent whole fish and the population estimates would be more comparable and relatable to other studies. In Table B), the population size seems to be standardized based on the area considered (is this hectares?). Please include units for area considered and population size (fish per ___). NOTE: same comments for Tables A) and B) apply to Tables C) and D).
Reviewer 2 Report
Labonne et al. have conducted a multi-generation reciprocal transplant study in natural introduced populations of trout to test for evidence of local adaptation and genetic rescue. The system in which this study was conducted, human introduced brown trout populations on a remote Antarctic island, is rather unique. The historical record of the introduced populations adds a level of detail that is rarely available for such studies. Given that the populations were intentionally founded, and isolated on a remote island, the introduction of a reciprocal transplant of fish in the natural environment seems appropriate, and received an ethical review and formal authorization. The major finding of the study is that the ‘local’ and ‘foreign’ individuals behaved quite differently in the two stream systems. In one stream, foreign introduced individuals contributed substantially to the gene pool in just two or three generations (high gene flow). In the other stream, the reciprocal foreign introduced individual contribution to the gene pool was comparatively much more modest (low realized gene flow). The authors argue against fitness components of local vs introgressed individuals resulting in signatures of local adaptation or genetic rescue, and instead argue in favor of sexual selection.
I think overall that the study was well done. I find the uniqueness of the experimental design very interesting. I have some questions about the results and conclusions that may either require further justification by the authors, or possibly clearer explanation.
Here I provide some notes on areas of the manuscript that I think either require further explanation, or with which I disagree:
Page 4, lines 144-146 – this is where the authors perform what I think is an all-important estimate of the population density of fish in each stream system. I believe there is not enough information provided for me to understand how the authors estimated population density. The authors refer to “supplementary material 1”. I don’t have supplementary material 1, but I think the authors are actually referring to Appendix C. Appendix C reports density estimates of various age classes, but not really how those densities were calculated. I think these density estimates are critical for the models, and I think the reader needs to know more about how they were calculated and assumptions that were made when they extrapolated to the entire population.
Page 5, lines 206-220 – this is where the authors describe how they used fish density (number of fish per surface area) to estimate the proportion of foreign individuals under two scenarios – a restricted movement/limited range scenario, and a total drainage area scenario. This is basically where the authors incorporate census population size into their models so they can know what portion of the C0 generation was migrant individuals from the other system. I think their models are critically dependent on this variable, and I would like to know more about how they estimated this parameter. For example, did the authors estimate fish density at one point in the drainage, and then assume a uniform density of fish across the entire surface area of the drainage? It seems like that is what they did, and that seems like it could be overly simplistic. The proportion of migrant individuals at C0 impacts expectations on the “percentage of transplanted individuals in each population” (table 1 legend). What if the authors’ percentage of transplanted individuals was way off due to incorrect assumptions about the uniformity of density of individuals across the entire drainage? How would that affect results reported in table 1? It could dramatically throw off their “expected fitness values”.
Page 6, lines 248-249 – I think these are instructions to authors, and need to be removed from the manuscript.
Page 7, Figure 2 – The y-axes are labeled as “frequency”, but the scales imply something like counts. I think these numbers match up to sample sizes reported in Appendix B table D.
Table 1 and Table 2 legends – can observed fitness be “total number of offspring sired” when it is based on a sample of individuals that were genotyped? Total number of offspring sired is a population parameter (not a sample parameter) that the authors cannot know.
Table 2 – I tried to understand how the authors came up with “expected fitness” values. On page 5, line 204 the authors say these numbers are based on pedigree data. I think pedigree data came from the analysis with the program COLONY, described on page 5, beginning on line 189. I’d like to know more about the assignment of individuals to families. How many families were identified and how big were the families? How much uncertainty was associated with assignment of individuals to families? How many adult spawning males and females are in the total population, and how many families would be inferred from a sample of several hundred individuals? What is the level of uncertainty associated with “expected fitness” in Table 2, and how might that uncertainty affect the reported goodness of fit tests?
Figure 3 – The selection of blue and green colors were hard to differentiate on my laser printer copy of the manuscript, and not really much better on the computer screen. Better contrasting colors might be easier to see. Green and red are not good colors to select for people who are red-green color blind.
Figure 2 – This figure seems to provide pretty compelling evidence that the two drainages responded differently to the introduction of equal numbers of transplant individuals. Although, maybe not, if the proportion of individuals who were foreign were way off relative to the percentages reported in Table 1. For example, if the Val Travers population was substantially smaller than the Claree population such that the proportion of foreign individuals was substantially higher, then the relative proportions of genotypic classes between the two drainage systems in both C1 (2012) and C2 (2018) look pretty similar to me.
The authors do a good job of discussing their results. However, I have enough questions about the methods and the results, that I don’t think I can critically evaluate the author conclusions regarding the local adaptation and genetic rescue hypotheses. However, the “4.3 Other drivers of gene flow” (page 13, starting on line 413) seems speculative to me, and not supported by the analyses. It seems like the argument is if it is not local adaptation, and it is not genetic rescue, then it must be sexual selection. To this point I think the final statement on page 14, line 452 – “our findings adds [sic] to the growing evidence that sexual selection may have tremendous effect on evolution” is highly speculative and over stated.
My two main reservations about this paper stem from a) my concerns about the population size estimates in the two systems, and the impact that these estimates can have on subsequent analyses as reported in table 1, and b) the lack of information about how the pedigree/family analysis was conducted with COLONY, the results of the pedigree analysis, and the importance of this analysis and its uncertainty to expected fitness values as reported in table 2.
Round 2
Reviewer 2 Report
I can see that the manuscript has been substantially edited between the original and revised versions. The quality of the science remains high, and the quality of the clarity of the writing has improved substantially. The authors did a good job of responding to my previous comments. The authors identified some misunderstandings I had with the analyses they conducted, and I think their explanations were good.
One thing I think I still disagree with is the useage of the word pedigree. I think this was a source of confusion for me in the first draft. The authors use the word pedigree interchangeably with hybrid index (line 138 on page 3). I don't think this is the appropriate usage of the term, and I don't think that Newhybrids can be used to construct pedigrees, which trace family relationships among individuals to each other. NewHybrids assigns hybrid classes (e.g. parental, F1, F2, first generation backcross, etc) to individuals, which is not the same thing as a pedigree. Individuals unrelated by pedigree can still belong to the same hybrid class. I would encourage revisions where this term is in use.
Author Response
We have modified the manuscript as suggested by Reviewer 2. We used "assignation to genotypic classes" or "structure of the gene flow" depending on the context, instead of "pedigree".